# Transcription Factors in the Pathogenesis of Lupus Nephritis and Their Targeted Therapy

**DOI:** 10.3390/ijms25021084

**Published:** 2024-01-16

**Authors:** Kasey M. Shao, Wen-Hai Shao

**Affiliations:** 1Department of Chemical and Biological Engineering, Princeton University, Princeton, NJ 08544, USA; 2Division of Rheumatology, Allergy and Immunology, Department of Internal Medicine, College of Medicine, University of Cincinnati, Cincinnati, OH 45267, USA

**Keywords:** lupus nephritis, transcription factor, targeted therapy

## Abstract

Systemic lupus erythematosus (SLE) is a prototype inflammatory autoimmune disease, characterized by breakdown of immunotolerance to self-antigens. Renal involvement, known as lupus nephritis (LN), is one of the leading causes of morbidity and a significant contributor to mortality in SLE. Despite current pathophysiological advances, further studies are needed to fully understand complex mechanisms underlying the development and progression of LN. Transcription factors (TFs) are proteins that regulate the expression of genes and play a crucial role in the development and progression of LN. The mechanisms of TF promoting or inhibiting gene expression are complex, and studies have just begun to reveal the pathological roles of TFs in LN. Understanding TFs in the pathogenesis of LN can provide valuable insights into this disease’s mechanisms and potentially lead to the development of targeted therapies for its management. This review will focus on recent findings on TFs in the pathogenesis of LN and newly developed TF-targeted therapy in renal inflammation.

## 1. Introduction

Systemic lupus erythematosus (SLE) is a multifactorial autoimmune disease that primarily affects females at the childbearing age [1]. Immune complex-mediated inflammation is responsible for clinical manifestations of SLE. Lupus nephritis (LN) is the most frequent and severe complication of SLE, accounting for the overall morbidity and mortality of the disease. About 50 to 70% of SLE patients will have renal inflammation and develop LN within the first 5 years of the diagnosis of lupus [2]. Current treatments for LN are ineffective and often toxic. Treatments typically involve immunosuppressive therapy, which is often associated with multiple comorbidities [3]. Despite advanced studies on the pathogenesis of LN, most clinical trials with targeted treatments have failed. Only three drugs (anifrolumab, voclosporin, and belimumab) have been approved for SLE and LN over the last 60 years. They often serve as an add-on therapy to the existing first-line immunosuppressants. Complete responses with advanced treatment are low, and relapses are common [4,5]. Therefore, there is an urgent need to uncover the immune mechanism and explore molecular pathway-targeting therapeutics for LN.

Transcription factors (TFs) are proteins that play a crucial role in gene expression. They are responsible for controlling the transcription of specific genes by binding to DNA and influencing the rate at which the information in those genes is converted into RNA [6]. This, in turn, impacts the production of proteins, which are essential for various cellular processes. TFs can activate or repress gene expression depending on the context and the specific genes they are interacting with [7]. They often work in coordination with other proteins to form complex regulatory networks that help cells respond to signals, environmental changes, and developmental cues [8]. As key regulators of immune cell differentiation and development, TFs have been associated with the pathogenesis of SLE and LN. Several TFs have been implicated in the development and progression of LN, and their dysregulation can lead to abnormal gene expression patterns, the activation of inflammatory pathways, and immune responses that contribute to kidney damage [9]. Understanding the role of TFs in the pathogenesis of LN is crucial for deciphering their regulatory mechanisms and developing targeted therapies that could help manage and treat this serious autoimmune kidney disease. Here, we will review some of the key TFs implicated in the pathogenesis and progression of LN. We will also review potential therapeutics with published data targeting those TFs.

## 2. TFs in the Pathogenesis of LN

### 2.1. STAT1 and IRF Are the Basis of the IFN-Signature

The interferon (IFN) signature, consisting of the increased transcription of both type I IFN and IFN-activated genes, is the dominant molecular pathway presented in cells from SLE patients [10]. The IFN signature also plays a key pathological role in pro-inflammatory cytokine secretion, immune cell infiltration and activation, and renal damage. Signal transducer and activator of transcription 1 and 2 (STAT1/2) and IFN regulatory factors (IRFs) are major TFs responsible for signature gene expression and effector phase execution (Figure 1). Plasmacytoid dendritic cells (pDCs) are the primary source of IFN-I in SLE patients, but mouse strains of spontaneous lupus do not appear to have an IFN signature, except pristane-induced lupus mice [11]. IFN-γ is considered a key mediator of LN in general, exerting disease-promoting effects at systemic and local axes in MRL/lpr and NZB/W F1 mice [12]. However, a study by Mohan’s group indicates that IFN-I produced by residential renal cells may contribute to end-organ damage through the recruitment of inflammatory monocytes and neutrophils [13]. The study was performed in C57BL/6J mice with anti-glomerular basement membrane (GBM)-induced nephritis [13]. Further investigation is apparently needed to validate the end-organ intrinsic IFN-I-exacerbated inflammation in spontaneous LN mice and in SLE patients. The activation of all three types of IFN receptors leads to the phosphorylation of STAT1, through the formation of homo- or hetero-dimers with STAT2. Therefore, STAT1 is the key molecule amplifying signature IFN genes [14]. The dysregulation of STAT activation has been associated with immune system dysfunction in LN. A differential analysis of transcription profile datasets from the NCBI GEO database found that STAT1 is closely related to LN [15]. Rahman’s group found that STAT1 serine-727 phosphorylation (STAT1-pS727) promotes autoimmune antibody-forming cell and germinal center (GC) responses, driving autoantibody production and LN development. However, STAT1-pS727 did not affect immune responses to the gut microbiota and foreign antigens [16]. Data implicate STAT1-pS727 as a potential therapeutic target. STAT1 expression and activation was found to be significantly increased in the kidney of MRL/lpr mice compared to sex/age-matched MRL/MpJ control mice [17]. Interestingly, negative feedback suppressor of cytokine signaling-1 (SOCS-1) and SOCS-3 are also significantly increased in the kidney of MRL/lpr mice [17] and in immune complex-induced glomerulonephritis [18]. We studied the reciprocal activation of STAT1 and STAT3 in renal inflammation in inducible lupus-like chronic graft-versus-host disease (cGVHD). STAT1 knockout (STAT1-KO) mice developed a significantly increased anti-dsDNA response, which was accompanied by increased proteinuria and a higher mortality rate [19]. Further investigation revealed increased STAT3 activity in the glomeruli of the STAT1-KO mice under cGVHD. Consistent with this, IL-6 and IFN-γ levels were significantly upregulated [19]. Treatment with the STAT3 Y705 inhibitor Natura-α in 19-week-old NZB/W F1 mice effectively reduced proteinuria, significantly improved survival, and reversed glomerular lesions [20]. Mice were treated for 29 weeks, and two doses (25 and 75 mg/kg) were given orally. Similar improvements were observed [20]. The findings from this study are especially encouraging because current treatments mainly focus on slowing down the deterioration of kidney injuries, and the reversal of glomerular lesions has not been achieved in LN patients. Consistent with this, STAT3 activity was reported to be significantly increased in glomerular cells from LN patients [21]. Several STAT3 small-molecule inhibitors (C188-9, OPB-31121, and analogs) with high potency are in clinical trials for treating advanced cancers. However, the therapeutic application of these compounds in LN has not been investigated.

STAT1 also forms heterodimers with IRF9 to form IFN-stimulated gene factor 3 (ISGF3), inducing IFN-stimulated genes from IFN-stimulated response elements (ISREs). In LN, abnormal interferon signaling and dysregulated IRFs have been linked to immune system hyperactivity and kidney inflammation. IRFs are involved in the pathogenesis of LN, and polymorphisms of IRF3, IRF5, and IRF7 are associated with LN [22,23]. MRL/lpr.IRF1-KO mice showed decreased inflammatory mediator production, decreased nephritis, and increased survival compared to MRL/lpr.IRF1-sufficient mice. The resulting IRF1 deficiency protection may be synergistically attributed to enhanced CD4^+^/CD25^+^ T_reg_ cells and decreased mesangial cell nitric oxide and IL-12 production [24]. IRF1 was identified as a direct target of microRNA (miR)-130b in mesangial cells in the kidney, and inhibiting IRF1 through miR-130b overexpression in NZB/W F1 lupus mice reduced IFN-accelerated nephritis progression, demonstrated by decreased proteinuria and glomerular lesions [25]. IRF5 siRNA-treated NZB/W F1 lupus mice showed a reduced anti-dsDNA response, decreased macrophage and T- and B-cell infiltration in renal tissue, and ameliorated renal disease outcomes compared with the non-treated group [26].

Despite the suppressive function of IRF4 in plasma levels of TNF and IL-12p40, IRF4 deficiency completely protected B6/lpr mice from glomerulonephritis [27]. B6/lpr.IRF4-KO mice are hypogammaglobulinemic and lack anti-nuclear and anti-dsDNA autoantibodies. The protected nephritis in these mice is probably due to a systemic absence of autoAbs, because data from the same group showed that IRF4 is required for plasma cell maturation [27].

### 2.2. NF-κB Is the Key Player in LN Development and Progression

Nuclear factor kappa B (NF-κB) regulates the expression of genes involved in inflammation and immune responses [28]. It governs the levels of cytokines/chemokines that mediate LN pathogenesis (Figure 1). Elevated NF-κB expression and activation have been documented in glomerular endothelial and mesangial cells in the kidneys of both lupus mice and SLE patients [29,30]. In SLE patients, NF-κB activation can lead to the production of pro-inflammatory cytokines (IL-6, TNF-α) and recruitment of immune cells (T cells and macrophages) to the kidneys, promoting kidney inflammation and damage [30]. Many gene variants along the TLR/NF-κB pathway have been identified in association with LN [31]. Both the canonical (TGFβ-activated kinase 1 (TAK1)-dependent) and non-canonical (NF-κB-inducing kinase (NIK)-dependent) activation pathways of NF-κB contribute to renal pathogenesis in LN [29,32]. Zheng et al. found extensive upregulation and activation of NF-κB in renal tubular cells and interstitial cells in SLE patients with nephritis compared to the controls [33]. The expression of activated NF-κB also correlated with renal disease activity. Data also indicated that canonical NF-κB activation contributes to tubulointerstitial lesions, while the non-canonical activation of NF-κB pathway is responsible for tubular cell damage [33]. Dehydroxymethyleposyquinomicin (DHMEQ, which inhibits NF-κB p65 subunit nuclear translocation) treatment reduced the number of renal lesions caused by pristane in BALB/C mice [34]. Two years later, similar results were demonstrated by Miyagawa et al. Injections of NF-κB inhibitor (SN50 peptide) into the C57BL/6 IRF7^−/−^ mice with LN induced by pristane markedly attenuated proteinuria and renal pathologic changes [35]. It is important to note that inhibitors targeting NF-κB upstream (TLRs, MyD88 et al.) and effector molecules (IFNs, IL-6) have been developed and approved to be active in diseases such as LN [31].

### 2.3. CEBPB Enhanced Inflammasome Activity in the Pathogenesis of LN

CCAAT/enhancer-binding protein beta (C/EBPB, or CEBPB), a leucine zipper TF, belongs to the CCAAT/enhancer-binding protein family. CEBPB is mainly involved in the differentiation of macrophages and granulocytes in the immune system [36]. It can also physically and functionally interact with members of the NF-κB family of TFs to induce sequential responses to inflammatory stimuli [37]. The importance of CEBPB was first reported by a study involving Chinese SLE patients. CEBPB mRNA expression was significantly elevated in PBMCs from a cohort of 20 SLE patients compared to 20 gender/age-matched healthy controls [38]. Further studies demonstrated that CEBPB expression was also significantly elevated in LN patients and possibly positively associated with disease progression [39,40]. Interestingly, higher-CEBPB SLE patients tend to have a high titer of certain autoAbs (anti-Sm, anti-SNP, or ANA). The levels of C/EBPB expression is also positively correlated to the SLEDAI score [38]. Pasula et al. later reported that CEBPB binding to the enhancer region of tumor necrosis factor alpha-inducible protein 3 (TNFAIP3, a negative regulator of pro-inflammatory stimulation) is required for its expression, while the TNFAIP3 locus has been shown to be associated with SLE by many studies [41]. Direct involvement of CEBPB in LN was discovered by Wang’s group [42]. They first showed that increased CEBPB expression is associated with nephritis progression in MRL/lpr mice. They then demonstrated that lentiviral shRNA-CEBPB mediated CEBPB knockdown in MRL/lpr LN mice reduced renal pathogenesis and improved renal function [42]. LN was similarly improved in MRL/lpr mice treated with the adeno-associated virus of CEBPB shRNA [39,40]. Collectively, studies suggest that three possible mechanisms are linked to the pathological role of CEBPB in LN: (1) CEBPB enhances NLRP3 inflammasome activation because the inhibition of CEBPB expression is associated with decreased NLRP3 levels [42]. (2) CEBPB is predominantly expressed in macrophages in the kidney. CEBPB-activated basic leucine zipper and W2 domain-containing protein 1 (BZW1) promotes endoplasmic reticulum stress-amplified macrophage inflammatory responses via eukaryotic initiation factor 2 alpha (elF2α) phosphorylation [39]. (3) CEBPB activates absent in melanoma 2 (AIM2, an IFN-inducible protein) inflammasome and podocyte pyroptosis by binding to the promoters of AIM2 and CASPASE1 to enhance their expression, and the knockdown of AIM2 or (and) caspase-1 reversed the effects of CEBPB overexpression [40].

### 2.4. Fli-1 Regulates Inflammatory Cytokine Expression and Immune Cell Infiltration into the Kidney

Friend leukemia integration 1 (Fli-1) is a transcription factor that belongs to the E26 transformation-specific (ETS) family of proteins. Fli-1 is expressed in fibroblasts, endothelial cells, and immune cells [43]. It plays a crucial role in the regulation of gene expression and is involved in various cellular processes, including cell proliferation, differentiation, apoptosis, and immune responses. Fli-1 has garnered significant interest due to its role in regulating immune system function and potential implications in diseases like LN. The abnormal expression of Fli-1 has been associated with the pathogenesis of SLE in both patients and a murine model of lupus [43]. In lupus, aberrant Fli-1 expression and activity have been reported in immune cells, including T cells and B cells. Increased levels of Fli-1 were reported in the lymphocytes of SLE patients, which was correlated with disease activity [44]. It is thought that the dysregulation of Fli-1 may contribute to the breakdown of immune tolerance and the production of autoantibodies that are characteristic of SLE. Fli-1 has also been implicated in the pathogenesis of LN. Abnormal Fli-1 expression in renal cells may contribute to the inflammation and tissue damage observed in LN.

Targeted disruption of Fli-1 results in embryonic death due to thrombocytopenia and inadequate vascular development [45]. Fli-1 transgenic overexpression leads to a high incidence of a progressive immunological renal disease, and mice ultimately die of renal failure caused by tubulointerstitial nephritis and immune complex glomerulonephritis [46]. The decreased expression of Fli-1 in mice (Fli-1^+/−^ mice) results in increased survival, significantly reduced autoAb production, and remarkably diminished proteinuria with decreased renal pathological scores in NZM2410 LN mice [47]. The decreased expression of Fli-1 in hematopoietic cells seems to be sufficient to convey protection in LN development in MRL/lpr mice [48]. The function of Fli-1 is context-dependent and can vary based on the cell type and its interacting partners. Collectively, mechanistic studies suggest that Fli-1 contributes to renal damage in LN via three primary pathways: (1) it promotes the differentiation of cytokine monocyte chemoattractant protein 1 (MCP-1) and inflammatory cytokine (IL-1, IL-6 et al.) expression in renal endothelial cells [49,50]; (2) it upregulates chemokine (CXCL13, CCL2, CCL4, and CCL5) expression and increases immune cell (T cells, B cells, and macrophages) infiltration [51,52,53]; and (3) it regulates IL-17A expression and Th17 cell infiltration in the kidney [50].

Inhibitors of Fli-1 (camptothecin (CPT) and topotecan (TPT)) markedly ameliorate LN in NZB/W F1 mice [54]. CPT and TPT treatment significantly reduced MCP-1 production in IFN-activated human renal endothelial and mesangial cells [54]. However, Fli-1 restoration data indicate that CPT-reduced MCP-1 is only partially dependent on Fli-1 inhibition, as CPT and TPT are also topoisomerase inhibitors. Frese’s group showed that the topoisomerase inhibitor irinotecan efficiently suppresses LN in the same strain, with a prolonged survival rate [55,56].

A deeper understanding of Fli-1’s precise role may offer potential insights into the disease mechanisms of LN and the development of targeted therapies. More research is needed to decipher the expression pattern of Fli-1 in renal residential cells under normal status versus inflammatory conditions. It is not clear if Fli-1-bearing renal residential cells functionally participate in affecting the severity of LN. Fibrosis in LN represents a terminal pathway of sustained immune-mediated injury and has been recognized as a marker of injury severity, a predictor of therapeutic response, and a prognostic factor of renal outcome in recent years [57]. The inhibition or haploid deficiency of Fli-1 potentially activates TGF-β, leading to tissue fibrosis [43,58]. The long-term inhibition of Fli-1 in the kidney helps to suppress ongoing inflammatory conditions but may enhance the fibrosis process. Investigation into the role of Fli-1 in fibrosis is critical to achieving effective utilization of these therapies in LN.

### 2.5. Sp1 Upregulates Key Protein Expression in LN

Sp1 is a member of the specificity protein (Sp) family of TFs, which are characterized by their ability to bind to GC-rich sequences (GC boxes) in the promoter regions of target genes. Sp1 is a well-known TF that plays a significant role in regulating gene expression in eukaryotic cells [59]. It was first discovered as a factor that binds to the GC-rich regions of the SV40 virus enhancer [59]. Subsequent work found that Sp1 is ubiquitously expressed across cell types and regulates a wide range of genes involved in diverse cellular processes, including cell cycle regulation, differentiation, apoptosis, angiogenesis, inflammatory signaling, and immune responses [60]. It can also interact with other TFs and co-regulators to either activate or repress gene expression, depending on the specific target gene and cellular context [59].

In LN, aberrant gene regulation by Sp1 may contribute to the dysregulation of immune responses and the production of pro-inflammatory cytokines, leading to kidney inflammation and damage. Significantly increased levels of Sp1 expression were reported in the kidney of a mouse renal interstitial fibrosis model [61] and MRL/lpr mice (Figure 2). In vitro and in vivo data suggest that Sp1 participates in renal fibrosis by promoting TGF-β expression in normal rat kidney NRK-49F fibroblast cells [62] and mesangial cells [63]. Sp1-promoted TGF-β expression is partially mediated through the upregulation of a long non-coding RNA, gm26669. Consistent with these findings, Sp1-targeted therapy prevents its translocation into the nucleus and ameliorates renal interstitial fibrosis [61]. Sole et al. characterized miRNAs from urinary exosomes and found Sp1 to be one of the common targets of all three miRNAs identified in their study [64]. Their immunohistochemistry data showed that Sp1 expression was increased in LN patients with low and mainly moderate chronicity indexes (CIs), but significantly decreased in the high-CI group compared to healthy controls [64]. The authors suggest that upregulation of Sp1 may promote the inflammatory process in the proliferative glomerulonephritis, but later stages of renal fibrosis may be independent of Sp1 [64]. Work from our group has determined that Sp1 is a primary driver of Axl receptor tyrosine kinase expression in the kidney of MRL/lpr mice ([65] and Figure 2). Axl-promoted mesangial proliferation contributes to renal inflammation [66,67,68]. Inhibiting Sp1 may represent a plausible therapeutic target in mouse models of LN, as the new Sp1 mithralog inhibitor EC-8042 shows pleiotropic activities and less toxicity in cancer treatment [69].

### 2.6. Other TFs Implicated in SLE Nephritis Pathophysiology

Upstream stimulatory factor 2 (USF2) belongs to the Myc family and forms dimers with DNA-binding domains [70]. The protein and mRNA levels of USF2 were reported to be significantly higher in the kidney tissue of MRL/lpr mice compared to the kidney tissue of C57BL/6 mice [71]. However, a better control for MRL/lpr mice would be MRL/MpJ mice. It has long been reported that the inhibition of USF2 protects renal mesangial cells from TGF-β-induced apoptosis [72]. Xie et al. showed that shRNA-mediated USF2 knockdown inhibits podocytes pyroptosis and ameliorates renal injury in MRL/lpr mice [71], but the data need to be further validated. The expression and function of USF2 in the kidneys of SLE nephritis patients has yet to be established.

Krüppel-like transcription factors (KLFs) were reported to participate in renal physiology involving glomerular filtration, tubulointerstitial inflammation, and renal fibrosis in human and rodents [73]. KLF5 is highly expressed in the renal tissues of MRL/lpr mice and in the peripheral blood of LN patients. The shRNA-mediated knockdown of KLF5 reduced renal fibrosis in MRL/lpr mice [74]. One possible mechanism is through the enhanced transcription of myxovirus resistance 1 (MX1), an interferon (IFN)-induced GTPase [74]. KLF15 interferes with the NF-κB inflammatory pathway by interacting with the co-activator P300 [75]. Though increased KLF15 and decreased NF-κB were shown to be correlated with improved renal pathology in an inducible mouse model of LN [76], there is no direct link. LN mice with KLF15 transgene expression may help to reveal their direct involvement in LN.

E2F is a TF in mesangial cells that regulates cell cycle progression and apoptosis through transactivation of the cell cycle regulatory genes proliferating-cell nuclear antigen (PCNA) and cdk2 kinase [77]. Dysregulated E2F1 activity has been implicated in the proliferation and survival of immune cells in LN [54,78]. Dzau and colleagues treated Thy1.1 glomerular nephritis rats with liposomes containing high-affinity E2F-binding oligonucleotides (which serve as a decoy, preventing endogenous E2F binding on target genes). The E2F decoy specifically inhibits the mRNA expression of PCNA and cdk2 in the kidneys. A significantly decreased number of glomerular cells and the attenuation of glomerular injury were observed in E2F decoy-treated rats compared to untreated rats [79].

Nuclear factor erythroid-related factor 2 (Nrf2) is a major regulator of the antioxidant response. An increased oxidative damage-accompanied Nrf2 response was reported in renal biopsies from LN patients [80]. Pristane-induced LN mice with Nrf2-deficiency developed more severe renal damage compared to Nrf2-sufficient LN mice [80]. Importantly, sulforaphane-induced Nrf2 expression reduces renal damage in pristane LN mice. The protective role of Nrf2 seems to operate by negatively regulating the NF-κB and TGF-β1 pathways [80].

Nuclear factor of activated T cells 5 (NFAT5), also known as tonicity-responsive enhancer-binding protein (TonEBP), is a key TF involved in regulating gene expression induced by osmotic stress [81]. Recent studies point to a role in the development and activation of immune cells, especially macrophages. NFAT5 induces the generation of Th17 cells and pro-inflammatory macrophages, contributing to autoimmune diseases [82]. Yoo et al. found that kidney resident and infiltrating immune cells display elevated NFAT5 expression in SLE patients with nephritis compared to control patients with basement membrane disease [83]. Pristane-lupus mice with myeloid lineage-depletion of NFAT5 failed to develop lupus and LN. NFAT5-dependent TLR activation on macrophages seemed to be responsible for the unresponsiveness in these lupus mice [83].

Hypoxia-inducible factor 1 (HIF-1) plays a key role in the metabolic reprogramming of immune cells. HIF-1α expression is upregulated in both the glomeruli and interstitium in SLE nephritis patients. Glomerular HIF-1α expression was associated with the severity of renal histopathology and clinical manifestations in LN patients from the same study [84]. In MRL/lpr mice, HIF-1α expression is also upregulated and correlated with renal injuries [84], and hypoxia-mediated renal damage is partially dependent on HIF-1-mediated metabolic reprogramming in infiltrating CD4^+^ and CD8^+^ T cells [85]. HIF-1 blockade inhibits infiltrating T cells in B6.Sle1.Yaa LN male mice. Most importantly, HIF-1 inhibition reverses renal cortical hypoxia and injury in 10–12-week-old female MRL/lpr mice to nearly the same degree as in control MRL/MpJ mice [85]. The reversible effect of HIF-1 blockade has great clinical therapeutic potential.

## 3. Conclusions

Understanding the role of TFs in the pathogenesis of LN can provide valuable insights into the disease’s mechanisms and potentially lead to the development of targeted therapies for its management (Table 1). As research in this field continues, we may uncover more specific details about the involvement of TFs in LN and its potential as a therapeutic target. TFs presenting altered activity offers a promising choice since they are pivotal points in signaling pathways, and therefore, their inhibition may block several routes involved in inflammation and tissue damage in LN. The pharmacological inhibition of TF signaling can be achieved by directly targeting or indirectly blocking upstream signaling elements and downstream effector molecules. Despite promising results for many inhibitors targeting TF upstream molecules, many have been discontinued after clinical testing due to toxicity and a lack of clinical activity because of compensatory mechanisms.

On the other hand, nephritis development in SLE patients has distinct stages characterized by pathophysiological molecular and cellular features. TFs play a critical role in regulating these responses. Therefore, targeting feature TFs at different stages of nephritis development is key to the successful management of the disease. Apparently, there is a lot that must be done to achieve this goal. Science has just started to reveal the regulation of the complex network of TF interactions. on the other hand, cytokine signaling is complex, involving both pro-inflammatory and anti-inflammatory effects. The analysis of cytokine-induced TF activation may provide additional insights into key steps of the complex cytokine regulation network. Therapeutic targets are evolving with findings in this field. We hope individualized diagnoses will help to identify specific TFs at different stages of SLE nephritis development.

## Figures and Tables

**Figure 1 ijms-25-01084-f001:**
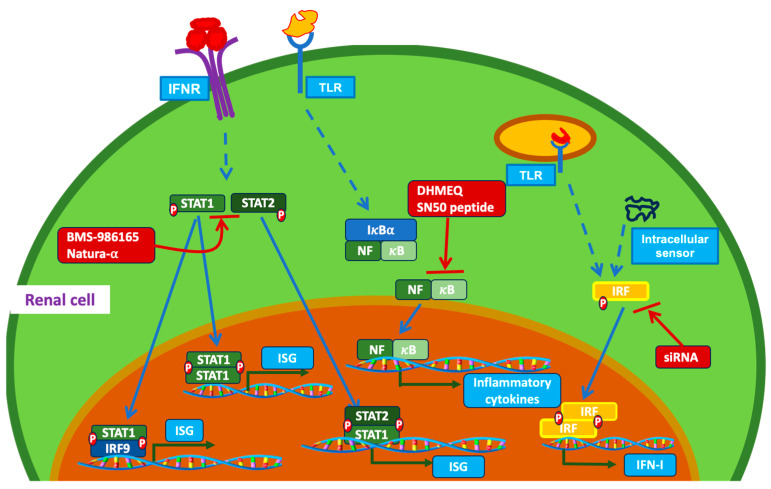
Three groups of major TFs involved in pathological damage in LN. Interferon receptor (IFNR) activation leads to STAT1/2 phosphorylation. Phosphorylated STAT1/2 forms STAT1 homomer or STAT1/2 heterodimer and translocates into nucleus to initiate IFN-stimulated gene (ISG) expression. STAT1 can also form heterodimers with interferon regulatory factor 9 (IRF9). Activation of cell surface TLRs initiates a signaling cascade to release NF-κB into nucleus to upregulate inflammatory cytokine expression and chemokine release. Upon binding to the ligands, intracellular nucleotide sensor and TLR activation results in IRF phosphorylation, which then translocates to the nucleus to start IFN expression/upregulation. Specific inhibitors to target these TFs are shown in red frames.

**Figure 2 ijms-25-01084-f002:**
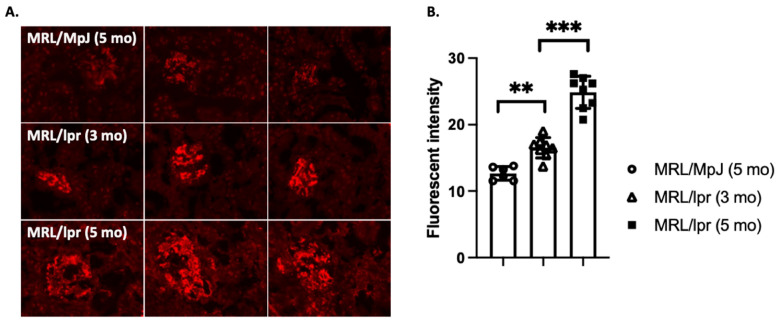
Increased Sp1 expression in the glomeruli of MRL/lpr mice. (**A**) Kidney sections (4 μm) were prepared from 3- and 5-month-old MRL/lpr mice and 5-month-old MRL/MpJ control mice and incubated with rabbit anti-mouse Sp1 antibody for 1 h at room temperature after blocking with TBST buffer containing 1% BSA. Sp1 expression was then visualized with PE-conjugated anti-rabbit secondary Ab. Images were taken under a Leica fluorescent microscope. (**B**) Sp1 expression levels were quantified via unpaired Mann–Whitney tests. Data are representative of three images from each of the 6 mice analyzed. ** *p* < 0.01; *** *p* < 0.001.

**Table 1 ijms-25-01084-t001:** Transcription Factors and Their Pathological Roles in Lupus Nephritis.

TFs	General Function	Pathogenesis in LN	Renal Cell Specificity	Inhibition in LN	Mouse Models
NF-κB	inflammation and inflammatory cytokines	pro-inflammatory cytokines (IL-6, TNF-α) immune cell infiltration	Glomerular endothelial, mesangial cells, tubular and interstitial cells	DHMEQ SN50 peptide	Balb/C Pristane lupus
STAT1/2/3	inflammation and inflammatory cytokines	monocytes and neutrophils infiltration	−	BMS-986165 Natura-α	MRL/lpr; NZB/W F1; B6 anti-GBM; B6 cGVHD-lupus
IRF3/4/5	Innate and adaptive immune responses	Macrophage/lymphocytes infiltration; anti-dsDNA	−	siRNA	MRL/lpr; NZB/W F1; B6/lpr
CEBPB	Macrophage/granulocyte differentiation	NLRP3 inflammasome AIM2 inflammasome Pyroptosis	Podocytes	siRNA/shRNA	MRL/lpr
Fli-1	Proliferation, differentiation, apoptosis	Chemokine/cytokines (IL-1, IL-6, CXCL13 et al); immune cell infiltration; IL-17A expression	Fibroblasts, endothelial cells	Camptothecin	NZM2410; MRL/lpr
Sp1	Cell cycle, differentiation, apoptosis	Renal fibrosis	Mesangial cells	Mithramycin EC-8042	MRL/lpr
USF2	Mitochondrial homeostasis,	Apoptosis, Pyroptosis	Mesangial cells Podocytes	siRNA	MRL/lpr
KLF	Metabolism	Glomerular infiltration Tubulointerstitial inflammation Renal fibrosis	−	shRNA	MRL/lpr
E2F	Cell cycle progression	Cell cycle Apoptosis	Mesangial cells	E2F decoy	Thy1.1 glomerular nephritis
Nrf2	Antioxidant response	Negative regulator of NF-κB and TGF-β1	−	−	Pristane lupus
NFAT5	Response to osmotic stress	Generation of Th17 cells and pro-inflammatory macrophages	−	−	Pristane lupus
HIF-1α	Metabolic reprogram	Renal cortical hypoxia	−	PX-478	MRL/lpr B6.sle1.Yaa

Note: NF-κB, nuclear factor kappa B; STAT, signal transducer and activator of transcription; IRF, interferon regulatory factor; CEBPB, CCAAT/enhancer-binding protein beta; Fli-1, friend leukemia integration 1; Sp1, specificity protein 1; USF2, upstream stimulatory factor 2; KLF, Krüppel-like transcription factor; E2F, E2 transcription factor; Nrf2, nuclear factor erythroid-related factor 2; NFAT5, nuclear factor of activated T cells 5; HIF-1α, hypoxia-inducible factor 1 alpha; DHMEQ, dehydroxymethyleposyquinomicin; NLRP3, NOD-like receptor protein 3; TGF-β, transforming growth factor-beta.

## Data Availability

Not applicable.

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
