# Peer review of "Transcription Factors in the Pathogenesis of Lupus Nephritis and Their Targeted Therapy"

_ijms, 2024, doi:10.3390/ijms25021084_

Round 1

Reviewer 1 Report

Comments and Suggestions for Authors

The authors reviewed several important transcription factors in lupus nephritis.

I have several suggestions to improve the quality of the manuscript and to make the contents more interesting to many readers.

How about making a table to summarize the important transcription factors?

I think the reasons why these transcription factors are being focused on, are weak.

I recommend discussing the association between the pathological classification of lupus nephritis and specific transcription factors. This emphasizes why these transcription factors are important and worthwhile to focus on.

I also recommend discussing the association between prognosis or response for the treatment and specific transcription factors.

Do you have any information on the new drug that targets these transcription factors the authors mentioned? If the authors have any information, ideally, the authors should list up and summarize it as a table.  

Author Response

How about making a table to summarize the important transcription factors?

 We thank the reviewer’s suggestion. We included a table in the revised manuscript.

I think the reasons why these transcription factors are being focused on, are weak.

We think there are strong evidence of these TFs in lupus nephritis. First, TF expression is significantly associated with LN disease progression in SLE patients; Second, TF upregulation promotes inflammation in LN as demonstrated by in vitro podocytes and macrophage studies and in vivo signaling pathway studies; Third, knockout or knockdown of TF gene expression in mouse models of LN resulted in improved renal function with pathological changes. We agree that there are currently no therapeutic drugs that directly target those TFs in LN patients.  

I recommend discussing the association between the pathological classification of lupus nephritis and specific transcription factors. This emphasizes why these transcription factors are important and worthwhile to focus on.

We thank reviewer’s suggestion. We have included pathological changes in mice lupus nephritis studies with specific TFs.

I also recommend discussing the association between prognosis or response for the treatment and specific transcription factors.

 There is no transcription factor-targeted treatment in lupus nephritis patients. TF-targeted treatment in LN mice are presented in Table 1 and discussed.

Do you have any information on the new drug that targets these transcription factors the authors mentioned? If the authors have any information, ideally, the authors should list up and summarize it as a table.  

We have summarized the targeted therapy in table 1.

Reviewer 2 Report

Comments and Suggestions for Authors

The article provides a correct summary of the role of transcription factors in the pathomechanism of LN. However, the article primarily lists the role of individual TFs without adequately situating and interpreting them as part of more general processes, resulting in fragmented individual chapters. However, the text fails to adequately highlight any specific molecular pathway that is dedicated to or perceived as such, despite presenting details about the role of TFs.
Are transcription factors involved, and if so, which ones, in LN formation by influencing autophagy?

Are transcription factors involved in the development of LN by influencing the inflammasome, and if so, which ones?

Concerning transcription factors, which epigenetic regulatory processes influence the development of LN?

To facilitate a concise description and illustrate the molecular relationships, we would need 2 or 3 figures. Each chapter presents a somewhat fragmented list of these relationships.

The number of references is quite short for a review. 

The manuscript is not suitable for publication in its current form. 

Author Response

The article provides a correct summary of the role of transcription factors in the pathomechanism of LN. However, the article primarily lists the role of individual TFs without adequately situating and interpreting them as part of more general processes, resulting in fragmented individual chapters. However, the text fails to adequately highlight any specific molecular pathway that is dedicated to or perceived as such, despite presenting details about the role of TFs.

We highlighted the dominant IFN pathway (section 2.1 and 2.2) in the pathogenesis of lupus nephritis. We now presented with a figure (figure 1) to illustrate this pathway and associated transcription factors.

Are transcription factors involved, and if so, which ones, in LN formation by influencing autophagy?

Autophagy in lupus nephritis has been reviewed by Podesta et al (Autoimmunity Reviews, 2022). It’s beyond the scope of the current manuscript.

Are transcription factors involved in the development of LN by influencing the inflammasome, and if so, which ones?

We thank the reviewer’s suggestion. We now included a new section (section 2.3.) to discuss the transcription factor CEBPB activated NLRP3 inflammasome and AIM2 inflammasome in LN.

Concerning transcription factors, which epigenetic regulatory processes influence the development of LN?

We thank the reviewer’s suggestion. Epigenetic regulation in lupus nephritis has been reviewed by others (Ning Xu, et al. Front Physiol, 2022; Xiaole Mei, et al. Kidney Diseases, 2022). Epigenetic alteration associated IRF and NF-kB regulation were discussed in those articles, but mainly related to immune cells, not directly to renal pathogenesis.

To facilitate a concise description and illustrate the molecular relationships, we would need 2 or 3 figures. Each chapter presents a somewhat fragmented list of these relationships.

We thank reviewer’s suggestion. We now included one figure and one table in the manuscript.

The number of references is quite short for a review. 

We understand that many review articles contain 100-150 citations. However, number of citations are often variable depending on the field to be reviewed. Transcription factors in lupus nephritis is a new field with ongoing studies. We only cite articles that are relevant in the manuscript. In addition, JAMA – The Journal of the American Medical Association requires review article to cite no more than 50-75 references. We now have 85 citations after the revision.

Round 2

Reviewer 1 Report

Comments and Suggestions for Authors

The authors addressed my concerns and added useful figure and table to the revised version.

Reviewer 2 Report

Comments and Suggestions for Authors

Thank you for the authors' consideration of my suggestions. The article has been properly and correctly revised. The new figures and tables are helpful for understanding. I consider the corrected version acceptable for publication.